# Routinely MUAC screening for severe acute malnutrition should consider the gender and age group bias in the Ethiopian non-emergency context

Masresha Tessema[1], Arnaud Laillou[2]*, Abiy Tefera[2], Yoseph Teklu[3], Jacques Berger[4], Frank T. Wieringa[4]

1 Ethiopia Public Health Institute, Addis Ababa, Ethiopia, 2 UNICEF Ethiopia, UNECA Compound, Addis Ababa, Ethiopia, 3 UNICEF Ethiopia, Emergency Nutrition Coordination Unit (ENCU), UNECA Compound, Addis Ababa, Ethiopia, 4 Institut de Recherche pour le Development (IRD), Montpellier, France

* alaillou@unicef.org

**Data Availability Statement:** All relevant data are within the manuscript and Supporting Information files.

## Abstract

Early identification of children <5 years with severe acute malnutrition (SAM) is a high priority to reduce child mortality and improved health outcomes. Current WHO guidelines for community screening for SAM recommend a Mid-Upper-Arm Circumference (MUAC) of less than 115 mm to identify children with SAM, but this cut-off does not identify a significant number of children with a weight-for-height Z-score <-3. To establish new specific MUAC cut-offs, pooled data was obtained for 25,755 children from 49 SMART recent surveys in Ethiopia (2016–2019). Sensitivity, proportion of false positive, and areas under receiver-operator characteristic curves (AUC) were calculated. MUAC below 115mm alone identified 55% of children with SAM identified with both methodologies. MUAC was worse in identifying older children (21%), those from a pastoral region (42%) and boys (41%). Using current WHO cut-offs, the sensitivity (Se) of MUAC below 115mm to identify the children severly malnourished screened through Weight-for-height below-3 was 16%. Analysing the ROC curve and Youden Index, Se and Specificity (Sp) were maximal at a MUAC < 133 mm cut-off to identify SAM (respectively Se 61.1%, Sp 81.4%). However, given the high proportion of false-positive children, according to gender, region and age groups, a cut-off around 125 mm to screen SAM could be the optimal one. In Ethiopia, implementation of a MUAC-only screening program for the identification of severe acute malnutrition with the actual cut-off of 115 mm would be unethical as it will lead to many children remaining undiagnosed and untreated. In addition, future study on implementation challenge on screening children with a higher cut-off or gender/age sensitive ones should be assessed with the collection of mortality and morbidity data to ensure that the most in need are being taking care of.

**Funding:** The author(s) received no specific funding for this work.

**Competing interests:** The authors have declared that no competing interests exist.

## Introduction

Ethiopia has experienced a rapid improvement on several nutrition indicators over the past fifteen years. However, the expected decrease in the number of stunted children from 5.85 to 5.11 million in 2025, corresponding to an average annual reduction rate (AARR) of 2.26%, is far from the projected AARR of 4.75%, needed to reach the 2025 World Health Assemble target [1]. In addition, over the last fifteen year, the prevalence of acute malnutrition has almost not changed (from 12% in 2000 to 10% in 2016) [2]. According to the 2019 Ethiopian Humanitarian Response Plan (HRP), almost two million children will be affected by acute malnutrition and will require assistance.

In Ethiopia, severe acute malnutrition (SAM), i.e. Mid Upper Arm Circumference (MUAC) <115 mm or weight-for-heigh z-score (WHZ) < -3 or nutritional oedema, affects approximately 1 million children under the age of five. Acute malnutrition places a child at a great risk for death and increases the risk of morbidity and stunting [3], therefore it is essential to prevent and treat acute malnutrition at an early stage. However, the 2025 World Health Assembly targets for Ethiopia will be challenging to achieve if current trends continue.

In the latest 2013 guideline for the management of severe acute malnutrition, the World Health Organization (WHO) recommends using Mid Upper Arm Circumference (MUAC) at community level to screen for SAM [4]. A recent meta-analysis published in 2018 [5] supports this recommendation that communities through different platforms could perform routine screenings of young children with MUAC and improve thereby detection of SAM cases. However, over the last years, several papers have highlighted that the current practice to identify children with acute malnutrition, at least in a non-emergency context, can be improved considerably by increasing the current WHO recommended cut-offs for MUAC [6–9]. In Cambodia [6] and India [9], the authors suggest that a cut-off towards 12.5cm to 12.8cm would increase the efficiency to screen children with severe acute malnutrition (SAM). Such change will allow to find with MUAC alone 50% of all the children with a limited proportion of false positive (<10%).

The present study aimed to answer several research questions posed in the revised WHO guidelines for the screening of severely malnourished children, especially focusing on the discrepancy between MUAC and WHZ, defining optimal MUAC cut-offs for routine system in non-emergency settings adapted to the Ethiopian context, and the influence of age and gender for detecting SAM in children under 5 years of age.

## Methods

We used pooled data from several SMART surveys carried out by the Government of Ethiopia and UNICEF, covering 4 years (2016–2019). Data on weight, height, MUAC, gender and age were available for a total of 26,806 Ethiopian children, from SMART surveys conducted in 49 different woredas/districts from 6 provinces of Ethiopia between 2016 and 2019. A two-stage cluster sampling approach based on a population sampling frame probability proportion to size (PPS) of all smallest rural villages was used as per SMART survey methodology. The sample size for anthropometric and retrospective mortality survey was determined by entering expected prevalence of malnutrition and mortality rate, desired precision, design effect, percentage of children under five, average household size and per cent of non-responsive households. Data quality was maintained by provision of standardization test and field practice during training, field level supervision by survey coordinators during data collection, daily checking and exchanging feedback on completed questionnaires and measurement error. The plausibility check in the ENA ensured that the quality of these data was within the expected quality standard.

All parents of participants gave their written consent. The target population was children aged 6–59 months. Children's height, measured to the nearest 1 mm (UNICEF measurement boards) were collected from each child. Weight was measured to the nearest 0.1 kg (Seca, Hamburg, Germany), with children wearing only light clothes. The nutritional status of children was defined using MUAC, length or height-for-age (L/HAZ), WHZ, and weight-for-age (WAZ) z-scores, calculated according to the Child Growth Standard of the WHO [10] using the WHO Anthro software. To ensure the accuracy of the data, extreme values were excluded from the analysis: WAZ < −6 or > 5; L/HAZ < −6 or >5; WHZ < −6 or > 5. Excluded values represented less than 4% of the total values (see Fig 1). The quality of the survey was also assessed through the z-score of WHZ (-0.615±1.048) as recommended by Grellety and al [11].

We used the following cut-offs to define acute malnutrition or wasting: WHZ < −2 Z-scores (according to the WHO growth charts) or MUAC < 125 mm, and severe acute malnutrition (SAM) was defined as: WHZ < −3 Z-scores or MUAC < 115 mm. Stunting was defined as a height-for-age < −2 Z-scores. The child's age in months was calculated by subtracting the date of the visit from the date of birth of the child; results were used as a continuous variable. Gender was considered as a binary variable.

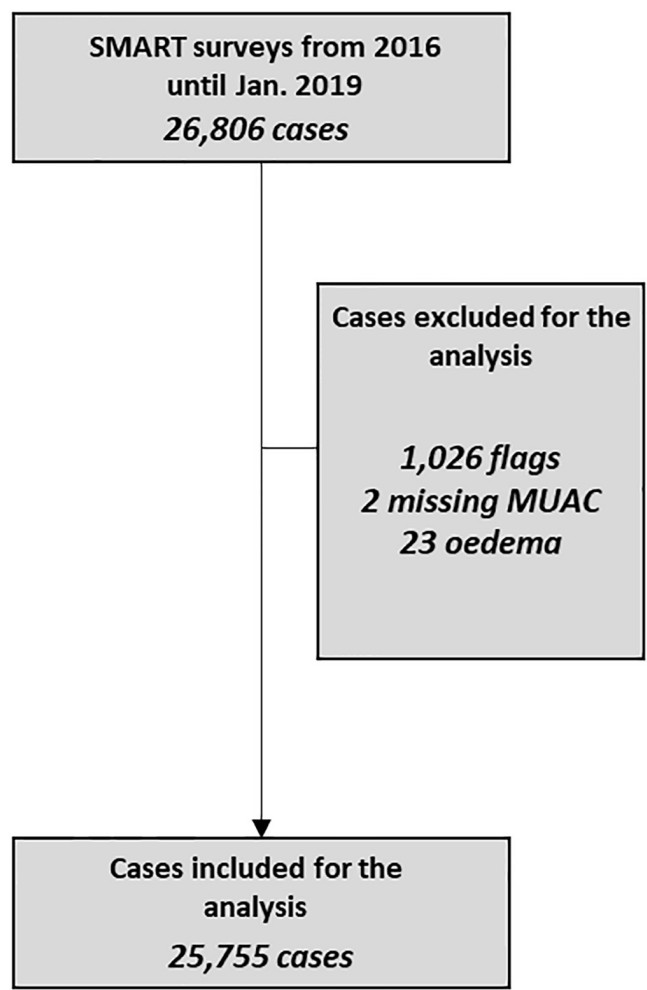

**Fig 1. Sample used for the analysis.**

We also investigated the potential bias of MUAC and WHZ as indicators for severe acute malnutrition. For SAM, a new variable with three categories was created: those with a MUAC<115 mm but a WHZ>-3; children with a WHZ<-3, but a MUAC>115mm; children with a WHZ<-3, and a MUAC<115mm with non-wasted children (WHZ>-3 and MUAC>115mm) used as reference category. For global acute malnutrition, similar categories were created using WHZ<-2 and MUAC<125mm. This variable was then used as a response variable to calculate sensitivity of each indicator to identify children with acute malnutrition or SAM.

To assess the performance of different MUAC cut-offs compared to the current cut-off recommended by WHO to detect severe and moderate acute malnutrition, receiver operating characteristic curves (ROC curves) were constructed. The sensitivity and proportion of false-positive (1-specificity) of MUAC were determined using wasting (WHZ<-2 z-score in children under 5 y) as acute malnutrition is the combined case load of low MUAC and/or low WHZ. The ROC curve is the plot of sensitivity versus proportion of false-positive of MUAC cut-offs. The area under curve (AUC) is the area between the curve and the segment (0,0) and (1,1), which corresponds to a random classifier. A larger AUC indicates a more accurate diagnosis of acute malnutrition defined by WHZ cut-offs [12]. Data analysis was performed using SPSS version 20.0 (SPSS, Inc., Chicago, IL).

To evaluate the performance of our analysis, the corresponding Youden index, which is the sum of sensitivity and specificity minus one, and reflects the overall capacity of an early warning model to detect a disease, was calculated: 1 indicating a perfect test, and 0 an imperfect test [13]. These analyses were conducted overall, and by gender, regions (agrarian versus pastoral) and age groups. Pastoralists are communities whose primary livelihood activity is livestock production/sales; whereas agrarian are those whose primary livelihood activity is crop production/sales, complemented with livestock production/sales. We used three criteria to which any new cut-off should adhere: i) Higher Youden index than the present cut-off; ii) Accuracy between 0.8 and 1; and iii) A proportion of false positive below 1/5 of non-malnourished children. The latter criteria is to prevent overburdening of health centres with too many false-positive cases of SAM. Calculations were done for all children overall, as well as for specific age groups, for girls, boys and region.

## Results

From the total of 25,755 children for whom data was available, 51% were male and 70.4% living in Agrarian zone. Mean (±SD) age was 31.7(±15.2) months (44% of children were less than 2 years old, 66% between 2 and 5 years old). Mean (±SD) MUAC was 141(±12) mm and ranged between 90 mm to 234 mm. Overall, 9.4% and 8.3% of the children were identified as GAM by WHZ<-2SD or MUAC<125mm respectively, while the prevalence of SAM (MUAC<115 mm or WHZ<-3) was identical for both indicators at 1.3%. The proportion of children indentified by both indicators was 19.2% for GAM and only 8.9% for SAM (Fig 2). Using MUAC-only identified 55.7% of the acute malnourished children and 55.2% of the severely malnourished children (Fig 2).

Subgroup analyses revealed large differences in the degree by which children were diagnosed by MUAC-only, ranging from 29.4% to 81.4% for GAM and from 21.4% to 75.1% for SAM, depending on the gender, region or age groups (Table 1). Less than half of malnourished boys and less than half of children living in the pastoral region were identified by MUAC. And MUAC identified less than ¼ of older children with SAM.

ROC curves (Fig 3) showed optimal MUAC cut-offs of 138 mm and 131 mm to detect also acute malnutrition and SAM identified by a low WHZ respectively. When MUAC <131 mm

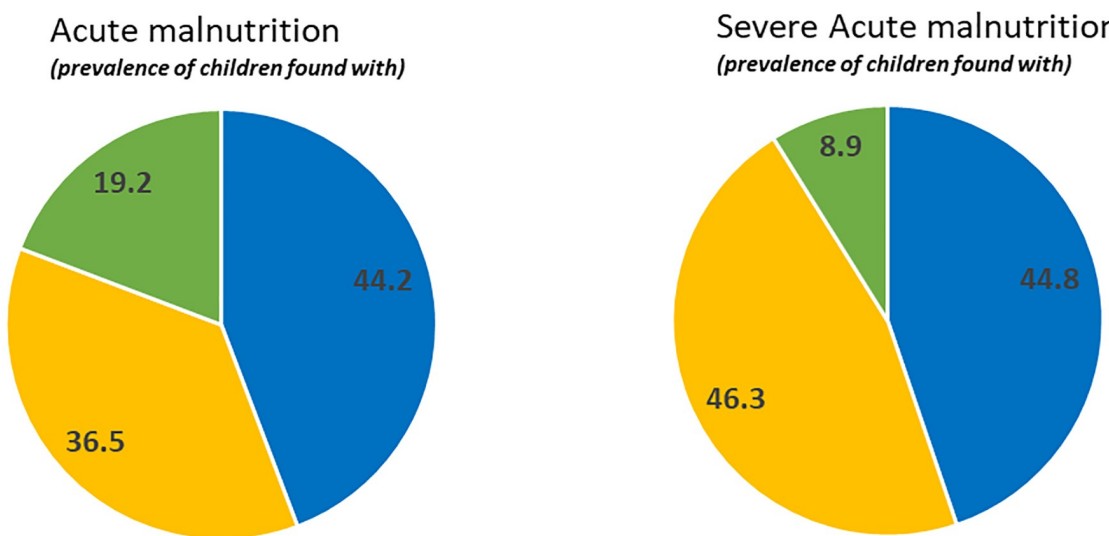

**Fig 2. Pie charts showing the proportion of children with GAM and SAM diagnosed by both MUAC < 125 mm and WHZ < -2SD (green) or by MUAC alone (yellow) or by WHZ alone (blue).**

is used to identify SAM, the sensitivity and specificity for our population surveyed reached respectively 61.1% and 81.7% with an accuracy of 0.814.

In Table 2, other MUAC cut-offs for detecting SAM are presented with consideration for other parameters than just the AOC (false positive, accuracy and Youden index). Increasing the MUAC to screen children for SAM from 115mm to 125mm could improve the sensitivity of the test by 2 to 10 times, depending on the sub-group assessed without overburdening the health post with too many false positive cases (less than 18%). Promoting a higher cut-off such as 131mm would continue to increase the sensitivity but the specificity would reduce significantly. For example, almost a fifth of the children screened will be false positive with a MUAC cut-off of 131 mm, while it is less than 8% for a cut-off of 125 mm. With the suggested cut-off of 125 mm, almost 30% more children would have been diagnosed as severely acute malnourished. For our complete data set, the number of children diagnosed with SAM would have increased 333 to 430 children.

**Table 1. Identification of global and severe acute malnutrition by weight-for height Z-score, mid-upper arm circumference, or by both criteria according to gender, age-group and age.**

| | GAM subject | WHZ<-2 only | MUAC<12.5 only | Both criteria | MUAC diagnosis | SAM subject | WHZ<-3 only | MUAC<11.5 only | Both criteria | MUAC diagnosis |
|---|---|---|---|---|---|---|---|---|---|---|
| *Gender* | | | | | | | | | | |
| Boy | 1,891 | 52.9 | 26.5 | 20.6 | 47.1 | 316 | 59.5 | 30.4 | 10.1 | 40.5 |
| Girl | 1,945 | 35.8 | 46.3 | 17.9 | 64.2 | 304 | 32.6 | 59.9 | 7.6 | 67.5 |
| *Age-group* | | | | | | | | | | |
| 6-23mo | 1,944 | 18.5 | 54.8 | 26.6 | 81.4 | 373 | 24.9 | 61.4 | 13.7 | 75.1 |
| 24-59mo | 1,892 | 70.7 | 17.8 | 11.6 | 29.4 | 247 | 78.5 | 19.8 | 1.6 | 21.4 |
| *region* | | | | | | | | | | |
| Pastoral | 1,392 | 57.7 | 24.8 | 17.5 | 42.3 | 246 | 58.5 | 31.3 | 10.2 | 41.5 |
| Agrarian | 2,444 | 36.6 | 43.2 | 20.2 | 63.4 | 374 | 38.2 | 53.7 | 8 | 61.7 |

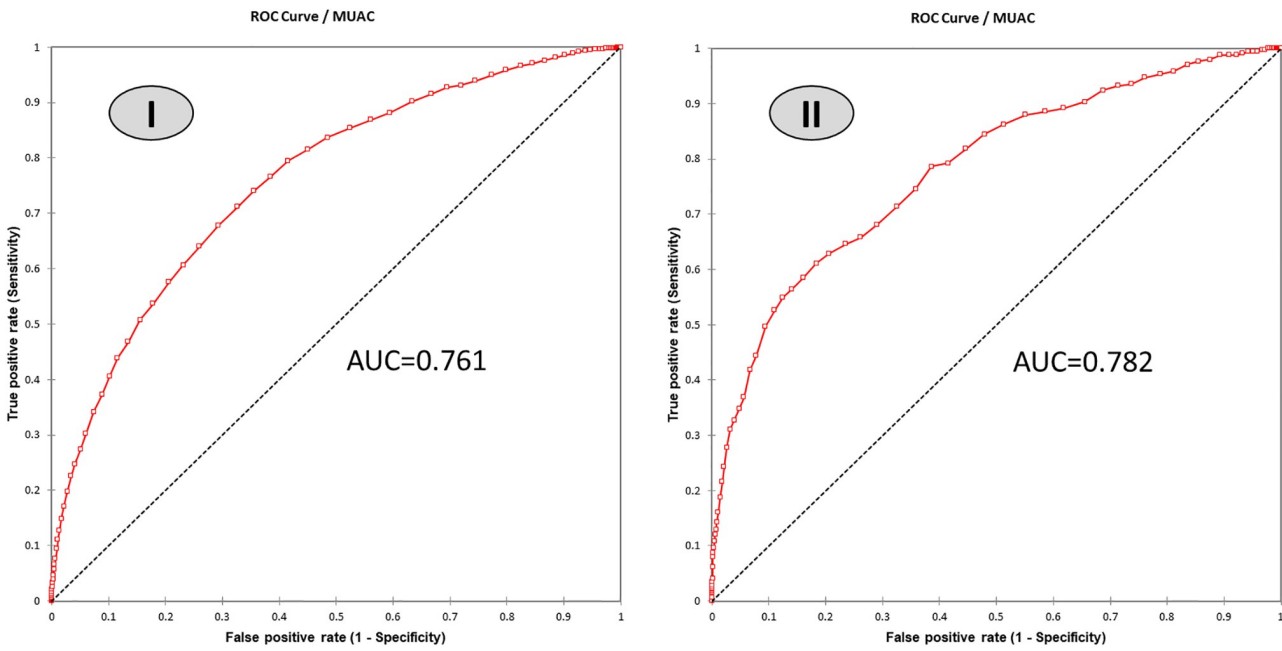

**Fig 3. ROC curve of the MUAC score against WHZ<-2SD (I) and WHZ<-3SD (II).**

## Discussion

It is accepted and highlighted in the WHO guidelines [14] that MUAC and WHZ indicators identify different categories of malnutrition and therefore different groups of acutely malnourished children. The present study confirms an earlier study from Southern Ethiopia [15] and studies in several other countries [9] that MUAC and WHZ identify different sub-sets of children. Also, our large sample from Ethiopia is highlighting similar findings from Tadess et al. [14] in that MUAC characterised a larger proportion of girls and young children as severely malnourished as compared to WHZ. In our study, about 60% of the boys and 79% of the older children failed to be identified as SAM through a routine community screening using current cut-offs. Those biases have been already described for Cambodia [7,16]. It is not surprising that MUAC is associated with age, as the current MUAC cut-off is age independent, while it is known that MUAC increases with age [16]. Hence, the older the child is, the less likely s/he will be detected as having SAM by a low MUAC. This bias of MUAC to detect younger children but also shorter ones was also already highlighted by other recent publication [17–18]. In most case, those children were considered at a higher risk of death [19–23] and responding well to treatment [9, 24–25]. However, according to a recent meta-analysis, there is increased risk of death with low WHZ as well [26].

Interestingly, a strong difference was also observed in our study between Pastoral and Agrarian regions, as MUAC failed to identify more children from Pastoral region (59%) than from Agrarian ones (38%). This might be due to body shape difference between regions measured by sitting-to-standing height ratio (SSR) as described by different authors [9,27] and even between different weather exposure [28–29]. In south Sudan, the SSR of patoralist population than settled one like agrarian population [9]. Difference in SSR might influence the diagnosis by WHZ independently of MUAC [9,27]. Therefore MUAC and WHZ are associated with different aspects of body composition, and therefore identify different groups of children with malnutrition [9].

**Table 2. Evaluation of screening test of nutritional status by different cut-offs of MUAC and WHZ (to detect severe acute malnutrition) in children aged 6–59 months.**

| | Sensitivity (%) | False Positive (%) | Accurarcy | Youden Index | Difference with highest Youden index (%) |
|---|---|---|---|---|---|
| *all sample* | | | | | |
| MUAC<11.5cm for WHZ<-3 (current practices) | 16.1 | 1 | 0.978 | 0.15 | -0.28 |
| MUAC<12.5cm for WHZ<-3 (most practical) | 44.4 | 7.8 | 0.915 | 0.37 | -0.06 |
| MUAC<12.7cm for WHZ<-3 (Considering other parameters) | 52.6 | 11 | 0.885 | 0.4 | -0.03 |
| MUAC<13.1cm for WHZ<-3 (optimal AUC) | 61.1 | 18.3 | 0.814 | 0.43 | 0 |
| *Gender* | | | | | |
| *boys* | | | | | |
| MUAC<11.5cm for WHZ<-3 (current practices) | 14.5 | 0.7 | 0.978 | 0.14 | -0.36 |
| MUAC<12.5cm for WHZ<-3 (most practical) | 45.9 | 6.2 | 0.93 | 0.4 | -0.1 |
| MUAC<13.1cm for WHZ<-3 (optimal AUC for all) | 65.9 | 16.2 | 0.835 | 0.5 | 0 |
| *girls* | | | | | |
| MUAC<11.5cm for WHZ<-3 (current practices) | 8.6 | 1.5 | 0.97 | 0.07 | -0.25 |
| MUAC<12.5cm for WHZ<-3 (most practical) | 28.2 | 9.5 | 0.894 | 0.19 | -0.13 |
| MUAC<13.1cm for WHZ<-3 (optimal AUC for all) | 48.2 | 20.3 | 0.791 | 0.28 | -0.04 |
| *Age group* | | | | | |
| *6–23 months* | | | | | |
| MUAC<11.5cm for WHZ<-3 (current practices) | 35.4 | 2.7 | 0.967 | 0.33 | -0.28 |
| MUAC<12.5cm for WHZ<-3 (most practical) | 77.8 | 17.2 | 0.827 | 0.61 | 0 |
| MUAC<13.1cm for WHZ<-3 (optimal AUC for all) | 84 | 25.9 | 0.644 | 0.58 | -0.03 |
| *24–59 months* | | | | | |
| MUAC<11.5cm for WHZ<-3 (current practices) | 2 | 0.3 | 0.986 | 0.02 | -0.4 |
| MUAC<12.5cm for WHZ<-3 (most practical) | 20.2 | 3.1 | 0.961 | 0.17 | -0.25 |
| MUAC<13.1cm for WHZ<-3 (optimal AUC for all) | 44.4 | 9.5 | 0.9 | 0.35 | -0.07 |
| *Region* | | | | | |
| *Pastoral* | | | | | |
| MUAC<11.5cm for WHZ<-3 (current practices) | 14.8 | 1 | 0.971 | 0.14 | -0.3 |
| MUAC<12.5cm for WHZ<-3 (most practical) | 41.4 | 7 | 0.919 | 0.34 | -0.1 |
| MUAC<13.1cm for WHZ<-3 (optimal AUC for all) | 61.5 | 17.2 | 0.824 | 0.44 | 0 |
| *Agrarian* | | | | | |
| MUAC<11.5cm for WHZ<-3 (current practices) | 17.3 | 1.1 | 0.988 | 0.16 | -0.27 |
| MUAC<12.5cm for WHZ<-3 (most practical) | 47.4 | 8.2 | 0.914 | 0.39 | -0.04 |
| MUAC<13.1cm for WHZ<-3 (optimal AUC for all) | 60.7 | 20.8 | 0.81 | 0.4 | -0.03 |

Age-dependent or perhaps even gender/region specific MUAC cut-offs would be more appropriate, however, it would need the development of such MUAC and additional training of the field. MUAC is clearly an easier and quicker screening tool at community level than weight-for-height. Mothers and health volunteers have shown their capacity of detecting SAM [5] in several countries of Africa with good sensitivity and accuracy [30,31]. Acute malnutrition screening should not be reduced to mortality risk screening but also to prevent morbidity, impaired physical, cognitive development, and associated micronutrient deficiencies [32]. Therefore, the cut-off of 115 mm is incorrect as single criteria to screen for severe acute malnutrition, as most children with a WHZ<-3 are missed. Increasing the MUAC cut-off for screening at 131mm as estimated by the ROC curve would increase significantly the sensitivity (from

16% to 61%), but at the same time, almost 26% of young children would be tested false positive for severe acute malnutrition. Acknowledging the potential and valid criticism of placing equal importance on Sensitivity and Specificity, a cut-off at 125mm seems the optimal one, as less than 8% of the screened children would be false positive and the sensitivity would move towards 44%, with an accuracy 91%. However, the suggested cut-off point for screening should further be studied in real practice or implementation, including measuring the additional burden to health system and cost-effectiveness in Ethiopia. It is important to start adcocating for targeted interventions to prevent wasting. Those false positive population could be one of them as they are towards the thine line of being considered as acute malnourished.

To avoid overburdening the Ethiopian health system with false-positive cases of severe acute malnutrition, we suggest including two MUAC cut-offs for different purposes. The first at MUAC<125mm could be used at community level to ensure inclusion of as many children with SAM as possible. This could identify, according to our data set, over 69% of the children with a WHZ<-3. Then, as a second step, all children with a MUAC below the screening cut-off (e.g., 125 mm) should be assessed at a health facility for weight, height and MUAC measurements following the WHO cut-offs which initiate treatment for severe acute malnutrition if SAM is confirmed and if not could initiate appropriate counselling or other programs to prevent future acute malnutrition problem. This approach may improve the cost-effectiveness of the screening programs and the treatment as the sensitivity will be significantly improved. Similar systems in other countries were proposed in other research [6,33].

Any gender analysis of acute malnutrition using the Ethiopian Demographic Health Survey (EDHS) should not undervalue the bias highlighted in this analysis. The EDHS only uses the weight-for-height methodology and therefore this would not identify the 46% of the children with a low MUAC-only, if the same trends are observed as in our study.

## Limitations of the study

A limit of the current study is that data used in this analysis was from none nationally represented surveys and therefore that the population screened by the different methods and the new MUAC cut-offs presented here are only adapted to the children assessed through those 3 years SMART surveys. Hence, the aim of the present study is not to suggest using the presented optimal cut-offs as international reference but as possibility for community screening in Ethiopia. However, the study highlights that in Ethiopia by using the actual cut-off of 115 mm results in almost 50% of children with SAM not receiving the vital treatment and/or preventing measures needed. Therefore, we recommend additional analysis in Ethiopia to develop a context wise community screening process to ensure that at least 80% of the children in need are correctly identified and treated.

## Conclusion

To conclude, the present study showed the ability of MUAC and WHZ to identify children with severe acute malnutrition. Both indicators showed gender, region and age bias. To ensure that no child with severe acute malnutrition is left without proper treatment and follow-up, a step-wise approach should be defined using MUAC<125mm used for community screening purpose and even the development of gender and age sensitive MUAC. However, further implementation studies need to better understand the health burden, the impact on morbidity/mortality and cost of additional screening.

## Supporting information

**S1 Data.**
(XLSM)

## Acknowledgments

We would like to thank Ethiopia ENCU, Nutrition cluster and their team for the data collection.

## Author Contributions

**Conceptualization:** Arnaud Laillou.

**Data curation:** Arnaud Laillou.

**Formal analysis:** Arnaud Laillou, Abiy Tefera, Frank T. Wieringa.

**Methodology:** Arnaud Laillou, Abiy Tefera, Frank T. Wieringa.

**Writing – original draft:** Masresha Tessema, Arnaud Laillou.

**Writing – review & editing:** Masresha Tessema, Arnaud Laillou, Abiy Tefera, Yoseph Teklu, Jacques Berger, Frank T. Wieringa.

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
