## [Decision Letter · Decision Letter 0]

29 Nov 2019

PONE-D-19-28795

Routinely MUAC screening for severe acute malnutrition should consider the gender and age group bias in the Ethiopian non-emergency context

PLOS ONE

Dear Dr Laillou,

Thank you for submitting your manuscript to PLOS ONE. After careful consideration, we feel that it has merit but does not fully meet PLOS ONE’s publication criteria as it currently stands. Therefore, we invite you to submit a revised version of the manuscript that addresses the points raised during the review process.

As the two authors clearly reported MUAC and WHZ identify different sub-sets of children with SAM and the two measurement should be regarded as independent from each other. So how it would be possible to assess the validity of MUAC based on WHZ cutoff values? Can WHZ be considered as a gold standard measurement for validating MUAC cutoff values? As provided below, the same serious concern has also been raised by both of the reviewers.It is not clear how the authors managed the data of children with nutritional oedema. Did you exclude or retained them in the analysis? I recommend to exclude them because, from practical perspectives, such children are automatically considered as SAM cases irrespective of their MUAC or WHZ measurements. Further, retaining them in the analysis may bias (overestimate) the WFH measurement of the children.The authors pooled data from several SMART surveys but said noting about the quality of arthrometric data of the original surveys. Did you exclude any SMART survey from the analysis due to quality concern? What were the quality assurance approaches employed by the individual surveys?The recommendation of the second author for validating MUAC against Age- and Sex-specify MUAC cutoffs should also be considered.

We would appreciate receiving your revised manuscript by Jan 13 2020 11:59PM. To enhance the reproducibility of your results, we recommend that if applicable you deposit your laboratory protocols in protocols.io, where a protocol can be assigned its own identifier (DOI) such that it can be cited independently in the future. For instructions see: http://journals.plos.org/plosone/s/submission-guidelines#loc-laboratory-protocols

We look forward to receiving your revised manuscript.

Kind regards,

Samson Gebremedhin, PhD

Academic Editor

PLOS ONE

Journal Requirements:

3. Please include additional information regarding the parental consent obtained for this study. For example, were the parents well informed prior to consenting?

5. Please remove your figures from within your manuscript file, leaving only the individual TIFF/EPS image files, uploaded separately.  These will be automatically included in the reviewers’ PDF.

Please format all the references according to the standard of the journal.

Additional (section-by-section) Editor Comments:

Background section

Line 63-66: can you please provide more details on the findings of the studies that recommended for increasing the current WHO recommended cut-offs for MUAC?

Methods section

Line 74-77: Please provide a brief highlight of the methodological approach of the SMART surveys included in the analysis.Can you please add a concise paragraph that provides an overview of Ethiopia including clear description of the pastoral and non-pastoral regions of the country?

Results

Please provide the proportion of children selected from the pastoral and agrarian regions.

Discussion

The authors concluded MUAC performs worse in identifying SAM children from a pastoral region of Ethiopia and justified that the finding can be due to “body shape difference” between Pastoral and Agrarian regions. But no explanation had been provided in what parameters the “body shape” is different between the two regions. Can you provide a brief highlight of the finding of the study by Myatt et al. [ ref # 18]?Line 171: (reference)????

Conclusion

Line 210 “Both indicators showed gender, region and age bias.” Did you identify any bias associated with the use of WHZ?

Reviewers' comments:

Reviewer's Responses to Questions

**Comments to the Author**

1. Is the manuscript technically sound, and do the data support the conclusions?

Reviewer #1: No

Reviewer #2: No

2. Has the statistical analysis been performed appropriately and rigorously? 

Reviewer #1: Yes

Reviewer #2: No

3. Have the authors made all data underlying the findings in their manuscript fully available?

Reviewer #1: No

Reviewer #2: Yes

4. Is the manuscript presented in an intelligible fashion and written in standard English?

Reviewer #1: Yes

Reviewer #2: Yes

5. Review Comments to the Author

Reviewer #1: This manuscript deals with an important topic.

The authors have made fundamental scientific mistakes by assuming weight for height as a gold standard and by trying to determine optimal cut offs by comparing the number of children detected by different anthropometric measures. The only way this can be achieved is by a study examining outcomes. Several studies have shown that weight for height and MUAC identify different groups of children and that MUAC has better predictive performance for mortality. Thus assuming weight for height as a gold standard is problematic because it has worse performance. The statement that switching to a MUAC based program would be unethical simply cannot be determined from the analysis that was done. It might be true, but it might also be true that a weight for height based program or a program based on both measures is unethical if it expends resources on a large number of children at low risk or it it fails to target those at greatest risk of death or neurodevelopment consequences. The other issues of ethical importance are coverage and cost benefit since unlimited resources will not be available and would need to be addressed in policy making decisions.

Overall the findings of how the two measures identify different populations in Ethiopia is of some interest in determining caseloads of potential different policies. The current WHO guidelines suggest using either measure but analysis of outcomes has led to a proposal that using a higher MUAC cutoff value would have better performance. Although the authors also suggest this, it should also only be based on a study that determines outcomes.

Reviewer #2: Routinely MUAC screening for severe acute malnutrition should consider the gender and age group bias in Ethiopian non-emergency context.

General comments.

The issue, of whether to use Weight-for-height or MUAC is still hotly debated, so far so that one has “believers” and “non-believers”.

This is mainly because there is never going to be a way to solve this issue. Firstly, what one is doing, is evaluating a screening tool to decide which one is best. This qualitative appraisal is done using sensitivity and specificity analyses and likelihood ratios to evaluate the text from a clinical perspective. The primary condition for this evaluation is that we “know” who is ill. Or to put it in another way ; that we know who is malnourished. Problem here is that we decide on this using an indicators that we are trying to evaluate. Here MUAC is evaluated against weight-for-height, or vice-versa. The flaw is that we are evaluating one indicator against another one without having a definition of who is really malnourished. We miss a “Golden Standard”.

The comparisons done in this present paper are therefore not very useful. Moreover, they say what many other papers have said already. Both indices find other populations and the overlap is small.

The second issue, is that arguments in favour of one or the other indicator are based on analyses of risk of mortality. Some studies find that MUAC is more predictive for mortality and hence should be used. The flaw here is that malnutrition is taken as an exposure variable, where it is rather an effect of a multi-causal problem. There a many different reasons why a child could be malnourished, and it is these reasons that are responsible for the increased mortality risk, and not the weight loss per sé.

Using malnutrition, based on an indicator, to predict mortality , is therefore not entirely correct from a strict epidemiological point of view.

So what can the authors do? They should have analysed more directly the effect of using a MUAC with single cut-off irrespective of gender. They should have done this using anthro and calculating the age and gender specific z-scores. Then they could have compared the misclassification comparing MUAC single cut-off with MUAC gender and age specific and distribution cut-off. This would have been more informative. The question of the study should have been whether to use MUAC with a single cut-off or use an age and gender specific one.

More specific comments

I have the impression that the authors have treated this “lightly” and not invested a lot in the existing literature. Moreover, on the HWO site is a useful systematic review on the matter: https://www.who.int/nutrition/publications/guidelines/updates_management_SAM_infantandchildren_review1.pdf

The paper would have benefitted from a more exhaustive literature evaluation.

Line 108. The Youden index is not correctly defined. It is not de difference between Se and SP; but rather Se+Sp minus 1.

How is accuracy measured and defined??

The rate in line 109 is not a rate, because the is no incidence and time line.

Table 2 is not useful , see my first main comment. The same goes for the AUC analyses.

Line 124: the figures are not corresponding to the figures.

Line 138. The false positive rates and the health post activities are not clearly explained. Is it because in the field MUAC is used , to be confirmed at the health post by Weight-for-height? In that case , indeed a lot of false positive will be identified. This is something to be avoided because it decrease the confidence the population will have in the health care providers. They go the health centre because they are told something is wrong with their child to be told late that there is no problem. The next time this happens, they will not go to the health centre again , because there previous experience thought them that nothing will happen. Late on, the authors use the higher sensitivity of MUAC to justify its use; forgetting the false positives at the health centre will have very important negative effects on health system appreciation. Line 187- 191

The discussion

This should be more focussed on the main question: MUAC one for all or a age and gender specific cut-off.

The argument of mortality should be re-evaluated against the second main comment I made.

Line 166: Why are logistic problem a challenge for MUAC???

Line 167: why is it a more “robust” screening tool???

Line 172: reference ?

Please also evaluated the quality of the surveys by giving an idea of the standard deviations as in the paper by Grellety. PLoS One. 2016 Dec 28;11(12):e0168585. doi: 10.1371/journal.pone.0168585. eCollection 2016. The Effect of Random Error on Diagnostic Accuracy Illustrated with the Anthropometric Diagnosis of Malnutrition. Grellety E1, Golden MH2.

6. PLOS authors have the option to publish the peer review history of their article (what does this mean?). If published, this will include your full peer review and any attached files.

Reviewer #1: Yes: James A Berkley

Reviewer #2: No

---

## [Author Response · Author response to Decision Letter 0]

27 Feb 2020

Editor Comments:

Background section

• Line 63-66: can you please provide more details on the findings of the studies that recommended for increasing the current WHO recommended cut-offs for MUAC?

We have summarized the main findings as requested.

Methods section

• Line 74-77: Please provide a brief highlight of the methodological approach of the SMART surveys included in the analysis.

Added as requested by the editor.

• Can you please add a concise paragraph that provides an overview of Ethiopia including clear description of the pastoral and non-pastoral regions of the country?

Added as requested for the agrarian and pastoral difference in the methodology section. We have also added an overview of the burden of acute malnutrition in Ethiopia in the introduction.

Results

• Please provide the proportion of children selected from the pastoral and agrarian regions.

As requested, we have added the # of children per area

Discussion

• The authors concluded MUAC performs worse in identifying SAM children from a pastoral region of Ethiopia and justified that the finding can be due to “body shape difference” between Pastoral and Agrarian regions. But no explanation had been provided in what parameters the “body shape” is different between the two regions. Can you provide a brief highlight of the finding of the study by Myatt et al. [ ref # 18]?

We have added a more substantial explanation on those factors which could influence body structure and therefore the use of MUAC or WHZ as a stand-alone.

• Line 171: (reference)????

Mistake, it has been taken out

Conclusion

• Line 210 “Both indicators showed gender, region and age bias.” Did you identify any bias associated with the use of WHZ?

We have included how WHZ is also associated to bias in the conclusion and that both methodologies are necessary

Reviewer 1:

The authors have made fundamental scientific mistakes by assuming weight for height as a gold standard and by trying to determine optimal cut offs by comparing the number of children detected by different anthropometric measures. The only way this can be achieved is by a study examining outcomes. Several studies have shown that weight for height and MUAC identify different groups of children and that MUAC has better predictive performance for mortality. Thus assuming weight for height as a gold standard is problematic because it has worse performance. The statement that switching to a MUAC based program would be unethical simply cannot be determined from the analysis that was done. It might be true, but it might also be true that a weight for height-based program or a program based on both measures is unethical if it expends resources on a large number of children at low risk or it it fails to target those at greatest risk of death or neurodevelopment consequences. The other issues of ethical importance are coverage and cost benefit since unlimited resources will not be available and would need to be addressed in policy making decisions.

We thank the reviewer for his comments, but do not agree with his statements. As we have tried to make clear in the manuscript is that cases of acute malnutrition consist of children with either a low weight-for-height, a low MUAC or both. There is no gold standard, and we have also not claimed that WHZ is superior of MUAC or vice versa. The purpose of our manuscript is to describe a way to identify as many children as possible with acute malnutrition. As MUAC is being used far more often in communes as screening tool than WHZ, is seems logic to try to optimize MUAC cut-off points to enable finding most children with also a low WHZ, without overburdening the health system. The statement from the reviewer “Several studies have shown that … MUAC has better predictive performance for mortality. Thus assuming weight for height as a gold standard is problematic because it has worse performance.” is wrong. The paper by Grettely and Golden [ref] shows clearly that WHZ is a better predictor of death than MUAC. Early papers were biased as children with both a low WHZ and MUAC were lumped together, creating a statistical artefact. Regardless of whether WHZ or MUAC predicts death better, children with a low WHZ or with a low MUAC are at an increased risk for death, something that has been shown by many studies. Treating only children with a low MUAC, and ignoring children with a low WHZ, is therefore clearly unethical. We have addressed the point on the coverage and the number of children falsely diagnosed in the discussion. 

Overall the findings of how the two measures identify different populations in Ethiopia is of some interest in determining caseloads of potential different policies. The current WHO guidelines suggest using either measure but analysis of outcomes has led to a proposal that using a higher MUAC cutoff value would have better performance. Although the authors also suggest this, it should also only be based on a study that determines outcomes.

In our opinion, the paper by Schwinger, Grellety, Golden, analyzing death as outcome in >15,000 children is conclusive and warrants treatment of children with severe acute malnutrition, regardless on the indicator used to identify the severe acute malnutrition. It serves nothing to endlessly repeat relatively small, under-powered studies on outcomes of severe acute malnutrition. 

Reviewer #2:

The issue, of whether to use Weight-for-height or MUAC is still hotly debated, so far so that one has “believers” and “non-believers”.

We do not categorize ourselves in Believers or not. We have just facts from Ethiopia. What do we do with the children not screened through MUAC alone. To date in Ethiopia, MUAC is used as the sole screening tool. 

This is mainly because there is never going to be a way to solve this issue. Firstly, what one is doing, is evaluating a screening tool to decide which one is best. This qualitative appraisal is done using sensitivity and specificity analyses and likelihood ratios to evaluate the text from a clinical perspective. The primary condition for this evaluation is that we “know” who is ill. Or to put it in another way ; that we know who is malnourished. Problem here is that we decide on this using an indicator that we are trying to evaluate. Here MUAC is evaluated against weight-for-height, or vice-versa. The flaw is that we are evaluating one indicator against another one without having a definition of who is really malnourished. We miss a “Golden Standard”.

As we have tried to make clear in the manuscript is that cases of acute malnutrition consist of children with either a low weight-for-height, a low MUAC or both. There is no gold standard, and we have also not claimed that WHZ is superior of MUAC or vice versa. The purpose of our manuscript is to describe a way to identify as many children as possible with acute malnutrition. As MUAC is being used far more often in communes as screening tool than WHZ, is seems logic to try to optimize MUAC cut-off points to enable finding most children with also a low WHZ, without overburdening the health system. As the condition ‘ill’, or in this case, acute malnutrition, is defined by a low WHZ and/or MUAC, theoretically, we would have captured 100% of cases with both indicators (which is different from for example an infectious disease, where you can have more negative indicators)

The comparisons done in this present paper are therefore not very useful. Moreover, they say what many other papers have said already. Both indices find other populations and the overlap is small.

See above. The main purpose of the study was not to show the lack of overlap (which has indeed been shown before), but how many more cases of severe acute malnutrition could be identified, and how this would affect case-load and false-positive rates.

The second issue is that arguments in favour of one or the other indicator are based on analyses of risk of mortality. Some studies find that MUAC is more predictive for mortality and hence should be used. The flaw here is that malnutrition is taken as an exposure variable, where it is rather an effect of a multi-causal problem. There a many different reasons why a child could be malnourished, and it is these reasons that are responsible for the increased mortality risk, and not the weight loss per sé.

Using malnutrition, based on an indicator, to predict mortality, is therefore not entirely correct from a strict epidemiological point of view.

We fully agree that there are many underlying causes of malnutrition, but the increased risk for mortality is one of the major reasons to treat malnutrition. The same can be said about malaria. There are many underlying reasons why a child becomes infected by malaria (lack of bednets, poor WASH with pools of water, etc), but the fact that the child is at an increased risk for death is reason for treatment. 

So what can the authors do? They should have analysed more directly the effect of using a MUAC with single cut-off irrespective of gender. They should have done this using anthro and calculating the age and gender specific z-scores. Then they could have compared the misclassification comparing MUAC single cut-off with MUAC gender and age specific and distribution cut-off. This would have been more informative. The question of the study should have been whether to use MUAC with a single cut-off or use an age and gender specific one.

This works has been done before, among others by us. See for example Fiorentino et al. that we have done in Cambodia. We do not see however how this would have addressed our prime objective of the study : based on current WHO guidelines for identifying (severe) acute malnutrition, how can assure that most children with (severe) acute malnutrition as diagnosed (and treated). 

The comparison of MUAC vs MUAC-for-age or MUAC-for-gender, leaves out the complete part of children diagnosed with WHZ alone, in our case, ~50% of children. We will concentrate another paper on this topics.

More specific comments

I have the impression that the authors have treated this “lightly” and not invested a lot in the existing literature. Moreover, on the HWO site is a useful systematic review on the matter: https://www.who.int/nutrition/publications/guidelines/updates_management_SAM_infantandchildren_review1.pdf

We do not agree that we have take this lightly, but we have included more literature as requested.

The paper would have benefitted from a more exhaustive literature evaluation.

We have added some more literatures into the paper.

Line 108. The Youden index is not correctly defined. It is not de difference between Se and SP; but rather Se+Sp minus 1.

Thanks for the reviewer, it was a mistake from our in the explanation. We have changed it and the table accordingly. 

How is accuracy measured and defined??

The percentage of biological implausible measures were very low. 3.8% of the values had a flag and were excluded from the analysis. As written, in the document, we have followed the recommended definition from WHO. To ensure the accuracy of the data, extreme values were excluded from the analysis: WAZ < −6 or > 5; L/HAZ < −6 or >5; WHZ < −6 or > 5. MUAC values were from 90cm to 200cm.

The rate in line 109 is not a rate, because the is no incidence and time line.

We have change it as suggested by the reviewer.

Table 2 is not useful, see my first main comment. The same goes for the AUC analyses.

See our comments previously as well, as we do think it is relevant.

Line 124: the figures are not corresponding to the figures.

What do you mean? If you ass for GAM 36.5+19.2=55.7% and for SAM 46.3+8.9=55.2…so yes they are corresponding to the figures.

Line 138. The false positive rates and the health post activities are not clearly explained. Is it because in the field MUAC is used , to be confirmed at the health post by Weight-for-height? In that case , indeed a lot of false positive will be identified. This is something to be avoided because it decrease the confidence the population will have in the health care providers. They go the health centre because they are told something is wrong with their child to be told late that there is no problem. The next time this happens, they will not go to the health centre again , because there previous experience thought them that nothing will happen. Late on, the authors use the higher sensitivity of MUAC to justify its use; forgetting the false positives at the health centre will have very important negative effects on health system appreciation. Line 187- 191

We do understand the concern, but if we are starting to work on the prevention of wasting, the false positive could be one of our focus as well. As they could cross the thin line which will categorize them as acute malnourished. We will add this notion in the text.

The discussion

This should be more focussed on the main question: MUAC one for all or a age and gender specific cut-off.

See our commented earlier. This works has been done before, among others by us. See for example Fiorentino et al. We do not see however how this would have addressed our prime objective of the study : based on current WHO guidelines for identifying (severe) acute malnutrition, how can assure that most children with (severe) acute malnutrition as diagnosed (and treated). The comparison of MUAC vs MUAC-for-age or MUAC-for-gender, leaves out the complete part of children diagnosed with WHZ alone, in our case, ~50% of children.

The argument of mortality should be re-evaluated against the second main comment I made.

Line 166: Why are logistic problem a challenge for MUAC???

We have erased this sentence as the only challenge is additional training and the development of the cut-off for MUAC gender and age based

Line 167: why is it a more “robust” screening tool???

Thanks to the reviewer, we have taken out the term of robust as it was out of line.

Line 172: reference ?

Thanks to the reviewer for highlighting this. We have resolved the issue as it was a mistake.

Please also evaluated the quality of the surveys by giving an idea of the standard deviations as in the paper by Grellety. PLoS One. 2016 Dec 28;11(12):e0168585. doi: 10.1371/journal.pone.0168585. eCollection 2016. The Effect of Random Error on Diagnostic Accuracy Illustrated with the Anthropometric Diagnosis of Malnutrition. Grellety E1, Golden MH2.

Included in the methodology as recommended. Our standard deviation for WHZ is pretty good as only 1.048SD.

---

## [Editor Report · Decision Letter 1]

3 Mar 2020

Routinely MUAC screening for severe acute malnutrition should consider the gender and age group bias in the Ethiopian non-emergency context

PONE-D-19-28795R1

Dear Dr. Laillou,

We are pleased to inform you that your manuscript has been judged scientifically suitable for publication and will be formally accepted for publication once it complies with all outstanding technical requirements.

With kind regards,

Samson Gebremedhin, PhD

Academic Editor

PLOS ONE
---

## [Editor Report · Acceptance letter]

20 Mar 2020

PONE-D-19-28795R1 

Routinely MUAC screening for severe acute malnutrition should consider the gender and age group bias in the Ethiopian non-emergency context 

Dear Dr. Laillou:

I am pleased to inform you that your manuscript has been deemed suitable for publication in PLOS ONE. Congratulations! Your manuscript is now with our production department. 

With kind regards,

on behalf of

Dr. Samson Gebremedhin 

Academic Editor

PLOS ONE